# Safety and Diagnostic Accuracy of the Transnasal Approach for Endobronchial Ultrasound-Guided Transbronchial Needle Aspiration (EBUS-TBNA)

**DOI:** 10.3390/diagnostics13081405

**Published:** 2023-04-13

**Authors:** Roberto Piro, Matteo Fontana, Eleonora Casalini, Laura Rossi, Maria Serena Simeone, Federica Ghinassi, Patrizia Ruggiero, Chiara Pollorsi, Sofia Taddei, Bianca Beghe’, Nicola Cosimo Facciolongo

**Affiliations:** 1Pulmonology Unit, Azienda Unità Sanitaria Locale—IRCCS di Reggio Emilia, 42123 Reggio Emilia, Italy; 2Respiratory Diseases Unit, Azienda Ospedaliero-Universitaria of Modena, 41121 Modena, Italy; 3Department of Surgery, Medicine, Dentistry and Morphological Sciences with Interest in Transplantation, Oncology and Regenerative Medicine, Faculty of Medicine and Surgery, University of Modena and Reggio Emilia, 41121 Modena, Italy

**Keywords:** endobronchial ultrasound-transbronchial needle aspiration (EBUS-TBNA), diagnostic accuracy, adverse events

## Abstract

Endobronchial ultrasound-guided transbronchial needle aspiration (EBUS-TBNA) is a safe and accurate diagnostic procedure used for investigating mediastinal pathologies. It is usually performed using an oral approach. The nasal route has been proposed but not extensively investigated. With the aim to report the use of linear EBUS through the nasal route and compare its accuracy and safety with the oral one, we conducted a retrospective analysis of the subjects who underwent an EBUS-TBNA procedure at our center. From January 2020 to December 2021, 464 subjects underwent an EBUS-TBNA, and in 417 patients, EBUS was performed through the nose or mouth. Nasal insertion of the EBUS bronchoscope was performed in 58.5% of the patients. No difference between the two insertion routes was observed in terms of location or number of stations sampled per subject. Procedure complications were mild and similar between the two groups (10.2% for the nasal group vs. 9.8% for the oral group). Minor epistaxis occurred in five subjects in the nasal group. Comparing the two groups, the rates of adequate specimens were similar (95.1% vs. 94.8%), as were the proportions of diagnostic specimens (84% vs. 82%). In conclusion, the nasal route for EBUS-TBNA is a valid alternative to the oral one.

## 1. Introduction

Endobronchial ultrasound-guided transbronchial needle aspiration (EBUS-TBNA) is an extremely safe [1] and accurate endoscopic procedure widely used for the diagnosis of mediastinal lymph nodes and lung lesions. This is a minimally invasive method to obtain samples for histopathological exams from mediastinal and hilar lymph nodes (2R, 2L, 3p, 4R, 4L, 7, 10R, 10L, 11R, and 11L) as well as lung parenchymal lesions. It is generally considered the first choice for sampling the mediastinum [2,3,4,5]. Two main types of EBUS probe are available: radial probe EBUS and linear probe EBUS. Whereas the first allows to visualize the wall layers and identify lung nodules, the latter is generally used for real-time sampling (EBUS-TBNA), as it can be attached to a transbronchial needle system [6,7,8].

The procedure can be performed under general anesthesia or deep sedation via endotracheal tube or laryngeal mask [9,10,11,12] or under moderate sedation with pharmacologically induced depression of the level of consciousness using benzodiazepines and opiates [13,14,15,16]. A randomized controlled trial by Casal et al. [13] showed no significant difference in diagnostic yield (70.7% vs. 68.9%, *p* = 0.816) and sensitivity (98.2% vs. 98.1%, *p* = 0.979) between the patients in the general anesthesia group and in the moderate sedation group. These results were confirmed by real-life studies, where EBUS-TBNA under moderate sedation proved to be feasible, maintaining adequate sampling and a high willingness to return [14,17]. It is worthy to consider that exams under deep sedation imply higher costs and that not all centers have constant availability of anesthesiologists; for these reasons, it is important to evaluate every possible way to optimize the patients’ tolerance for the exam and the endoscopist’s satisfaction during the procedures when the exam has to be performed under conscious sedation.

While for deep sedation the transoral approach is the only access modality, for moderate sedation the EBUS-TBNA procedure can be performed both through the nose and through the mouth [18]. The oral route is the traditional one, and it is the most used. On the other hand, nasal insertion, although well established for conventional bronchoscopy [19], has started to gain importance for EBUS-TBNA during the last few years [20,21,22], since EBUS bronchoscopes with a smaller tip are available. Indeed, a retrospective study [8] shows that EBUS-TBNA is possible through the nose in 73.5% of the patients, with no significant difference regarding the sampling, the procedure duration, or the complications compared with the oral route. Moreover, a randomized controlled trial [14] shows that the two approaches have no significant difference in patient comfort, satisfaction, and willingness to return, as well as in stations sampled, procedure duration, and total doses of sedatives administered. The rate of procedures in which the nasal insertion failed was 24.5%.

Considering the similar outcomes emerging from these initial observations, more studies are necessary to understand if the nasal route could represent a favorable alternative for EBUS-TBNA. Therefore, we conducted a retrospective monocentric cohort study to describe the diagnostic yield and safety of EBUS-TBNA via the nasal route compared to the oral route.

## 2. Materials and Methods

This is a retrospective, monocentric study conducted on patients who underwent endobronchial ultrasound-guided transbronchial needle aspiration (EBUS-TBNA) at the Interventional Pulmonology Unit of Reggio Emilia, a third-level center in Italy. We enrolled patients from 1 January 2020, to 31 December 2021. The aim of the study was to evaluate the outcome of the nasal insertion route for linear EBUS and compare it with oral access. This study has been approved by the Institutional Review Board of the “Area Vasta Emilia Nord” (authorization number 2022/0139556 of 11 November 2022). Data analysis was performed in accordance with the Declaration of Helsinki. When it was possible, a written informed consent to participate in the study was obtained from every patient. In consideration of the features of the retrospective study, the Institutional Review Board authorized the analysis of the data related to the patient who was not reachable to ask about consent. Privacy and anonymity were ensured for unreachable patients. 

The study was performed using the informatics database that collects all the data about the interventional procedures conducted at our hospital.

The patients included in the study had a clinical and radiological indication for EBUS-TBNA, and they had no contraindication to bronchoscopy and/or sedation. Every patient gave written informed consent to the procedure. Inclusion criteria also required an age greater than 18 years. EBUS-TBNA procedures performed with the use of a rigid bronchoscope were excluded from the analysis, as were intubated or tracheostomized patients. Subjects presenting with severe hemodynamic instability were not considered clinically eligible [23]. Subjects taking anticoagulant or antiplatelet agents (other than aspirin) had to withhold the medication according to the standard recommended period [24,25]. If this interruption was judged clinically contraindicated, the patient was ruled out of the study. The complete list of the inclusion and exclusion criteria is reported in Table 1.

The first analysis has been performed on subjects who underwent EBUS via the nasal or oral route; a second analysis has been performed taking into account patients who underwent EBUS with deep sedation via the laryngeal mask. In Figure 1, the different access routes and relative bronchoscopist positions are shown. Specifically describing the technique of access, the EBUS scope was introduced through the open nostril after using lubricating jelly. The scope was then gently introduced through the middle and lower nasal choana, without resistance and trying to avoid trauma to the nasal walls. The oral route was performed using a dedicated plastic mouthpiece, with the aim of avoiding scope damages, fixed between the dental arches and behind the head of the patients with a silicon lace. Finally, regarding the laryngeal mask route, it was adopted only with anesthesiologist assistance, after deep sedation and the introduction by the anesthesiologist himself of a new generation laryngeal mask (i-Gel, Intersurgical, Mirandola, Italy). Laryngeal mask is a silicon device consisting of a tubular part through which the bronchoscope is introduced and an oval-shaped distal end that is positioned above the laryngeal inlet, enveloping the epiglottis and arytenoids and allowing good ventilation and comfortable access to the laryngeal inlet without the need for curarization (as for the endotracheal tube placement) and permitting a better visualization of all mediastinal lymph node stations, including the upper paratracheal ones.

All interventional pulmonology procedures considered in the study were performed with an EBUS bronchoscope that has an outer diameter of 6,9 mm (Olympus BF-UC180F, Tokyo, Japan) or 6,6 mm (Olympus BF-UC190F). EBUS-TBNA procedures were performed by experienced interventional pulmonologists supported by two nurses specialized in this field. In some cases, a balloon device (Olympus) filled with sterile saline water was positioned at the beginning or during the procedure, depending on the preferences of the endoscopist. The drugs for conscious sedation and their dose were decided by the bronchoscopist following current clinical practice and according to the patient’s features. They were chosen between meperidine, generally administered as 1 mg/kg at the beginning of the procedure; midazolam, administered in small boluses of 1–2 mg, up to 15 mg total; and fentanyl, administered in small boluses of 0.025–0.05 mg, up to 0.2 mg total [14,18]. In the procedures conducted under anesthesiologist assistance, midazolam, fentanyl, remifentanil, and propofol have been used. All patients received local anesthesia in each nostril, in the oropharynx, in the larynx, and in the lower airways; in the latter, a “spray-as-you-go” technique was applied. The lowest dose of lidocaine to ensure good bronchoscopic conditions and patient comfort was used; approximately 4–6 mg/kg of lidocaine were administered. [23]. Usually in the same session, patients undergo a flexible bronchoscopy in order to examine the tracheobronchial tree before the EBUS-TBNA procedure. Explorative bronchoscopy was avoided only if it had already been performed in the days before.

The attempt to insert the instrument through the nose was first made based on the preference of the bronchoscopist. The patients were asked if a nostril was preferred for the passage of the instrument due to anatomical or anamnestical reasons, if not previously indicated by the patients themselves. In case of failure due to a narrowed nasal passage or the onset of either uncontrolled pain or bleeding, the procedure was carried out by inserting the EBUS bronchoscope through the mouth after placing a specific mouthpiece to prevent bronchoscope damage. A recent history of epistaxis was taken into consideration by the bronchoscopist when deciding the force to apply before considering the nasal approach. Both in the cases of nasal and oral access, the patient received oxygen through a nasal canula and underwent continuous multiparameter monitoring (saturation, electrocardiogram, and regular blood pressure measurements) [23].

Usually, during EBUS-TBNA procedures, at least three passes were performed at each lymph node station. In some cases, rapid on-site cytologic examination (ROSE) was performed [18,26], depending on the availability of a second pulmonologist or a biologist trained in that technique. The procedure planning (location and number of lymph node stations to sample) was determined case by case by the bronchoscopist based on the imaging and the diagnostic purposes for each patient. After the procedure was completed, outpatients were monitored for a period of at least two hours and then discharged, while inpatients returned to their respective hospital wards. In the event of complications such as bleeding or pneumomediastinum occurring within a few hours after the procedure, the bronchoscopist was consulted. Otherwise, in cases of longer duration, patients were usually referred to the emergency department.

The primary outcome of the study was to compare nasal and oral insertion routes in terms of the percentage of procedures that were successfully completed. Secondary outcomes included the analysis of performance elements, evaluated both on patients and procedures. The following variables were assessed for each patient: age, sex, weight, height, and BMI (body mass index); death at the time of data collection; previous biopsies performed; adenopathies present at imaging; the need for mediastinal EBUS staging; diagnostic yield and adequacy; amount of sedatives used; and the rate at which complications occurred during the procedure or within 24 h. The following variables were assessed for each procedure: number of lymph node stations sampled, percentage of adequate TBNA specimens, types and percentage of confirmed diagnoses, and number of passes for each lymph node station. The software used for the statistical analysis is Prism 9.5 for MacOS (GraphPad Software, San Diego, CA, USA, www.graphpad.com, accessed on 25 February 2023). The mean and standard deviation have been used to describe continuous variables, while categorical variables have been expressed with absolute values and percentages. *T*-tests with Welch correction or Mann-Whitney tests have been used to compare two groups of continuous variables, while Brown-Forsythe and Welch ANOVA tests and Kruskal-Wallis tests have been used to compare three groups of continuous variables. Categorical variables have been compared with the Chi-square test or Fisher’s exact test, where appropriate. A *p* value of less than 0.05 has been considered statistically significant.

## 3. Results

Between January 2020 and December 2021, 464 eligible patients underwent a linear EBUS at our center; 244 (52.6%) procedures have been performed via the nasal route and 173 (36.3%) via the oral route; of these procedures, respectively, 2 (0.4% of the total) and 7 (1.5%) have been performed with anesthesiologist assistance. The remaining 47 procedures (10.1%) have been performed via laryngeal mask (i-gel), all under anesthesiologist assistance. The division of the casuistry is shown in Figure 2.

The age, survival status, body features, data regarding samples, and results for the overall population are shown in Table 2.

At the comparison, the patients in the nasal access group and those in the oral access group showed differences in particular regarding sex and body size: the nasal route was more frequently adopted in male patients (69.3% vs. 55.5%), making them consequently likely to be taller and heavier (168.1 ± 10.3 cm vs. 165.4 ± 9.2 cm, and 74.8 ± 16.4 kg vs. 71.7 ± 16.8 kg). No substantial intraprocedural differences between the two groups have been reported, in particular regarding the adequate visualization of nodal stations, the performance of the biopsy, and the number of sampled nodal stations, as well as the number of total passes. Almost every nodal station has been considered for sampling and effectively sampled during procedures, even station 12L, which is rarely reachable by EBUS scope; 12R is the only station that has never been reached in our case study. Only for the sampling at station 2R we noticed a significant difference (2.9% vs. 0%, *p* = 0.045). A slight but significant difference in the percentage of patients who underwent systematic mediastinal staging has been found (30.1% vs. 21.4%, *p* = 0.04). Complications were defined as events that caused an interruption of the procedure or a need to administer more sedatives or other drugs in order to be resolved; they were overall infrequent (in particular represented by epistaxis, bronchial bleeding, desaturation, cough and stridor, or bronchospasm) and did not differ between the groups (10.2% for the nasal group vs. 9.8% for the oral group, *p* > 0.9). Especially, five patients developed epistaxis in the nasal group; the episode was minor and required no intervention except for tamponade, defined as the placement of a small, sterile gauze pad in the nostril where the epistaxis was observed. When stridor or bronchospasm were clinically detected, intravenous steroid therapy (usually methylprednisolone 40 mg) and aerosol therapy (usually with beta-2 agonists, antimuscarinic agents, and inhaled steroids) were administered during the procedure with rapid recovery of the objective signs. The rates of diagnostic EBUS-TBNA sampling were similar between the two groups (84 vs. 82%, respectively; *p* = 0.59). The sample was defined as inadequate in 4.9% of the nasal group procedures and in 5.2% of the oral group, while false negative cases have been identified after retrospective review of the cases for only two patients, both in the nasal access group. The doses of midazolam, meperidine, and fentanyl were comparable, and the same was true for remifentanil and propofol in procedures performed under anesthesiologist assistance; these were slightly, but significantly, higher for the oral group (2 vs. 7, *p* = 0.037). Data comparison for the two groups is shown in Table 3.

Considering the patients in which EBUS-TBNA has been performed in deep sedation through a laryngeal mask, we have not found differences in diagnostic accuracy and sample adequacy. A significant difference is present when considering patients who underwent previous biopsies (*p* = 0.02). Interestingly, we have also found that the complication rate (10.6%) was similar to that of the other groups. The most significant difference, apart from the sedative drug used, is related to the percentage of mediastinal staging performed with a laryngeal mask (53.2%, *p* = 0.0001) and, consequently, the number of stations sampled and total needle passes (2.1 ± 1.3 with a *p* = 0.02 and 7.4 ± 3.2 with a *p* = 0.0003, respectively). Data from the comparison between all three groups are presented in Table 4. For a complete comparison of the three groups on all variables analyzed, see Appendix A.

## 4. Discussion

Despite the fact that flexible bronchoscopies are generally performed through the nasal route, the oral route is the conventional approach for endobronchial ultrasound-guided transbronchial needle aspiration (EBUS-TBNA), in particular because of the larger scope size. Most operators are trained to execute EBUS-TBNA only through the mouth. On the contrary, the nasal route could provide more scope stability, leading to better performance of bronchoscopic procedures. Moreover, in the last year, new EBUS bronchoscopes have been available, with a compact and smaller distal tip that could pass easier in thinner nostrils. The choice of the oral/nasal approach is strictly linked with the sedation protocol applied. While the procedures under general anesthesia or deep sedation require the first, if the bronchoscopy is performed under conscious sedation, both are possible. Randomized controlled trials and retrospective studies [13,14] show no significant difference concerning the sedation protocol in diagnostic yield and sensitivity. For this reason and because of the scarce availability of anesthesiologists in some centers, conscious sedation is widely used for EBUS-TBNA, and the choice of the trans-nasal approach could gain interest because the evidence regarding the optimal method for performing EBUS is insufficient.

As already described for conventional bronchoscopy, the nasal insertion causes fewer retching reflexes [27], grants better control of the bronchoscope [28], and provides the opportunity to explore the upper airways [29]. Oral insertion is less dependent on the patient’s anatomy, and in a randomized clinical trial involving 66 patients [30], it has shown a significantly shorter time to pass the vocal cords. Notably, in the same trial, similar tolerability and willingness to return were reported between the two methods. As expected, a conversion to the oral route was needed in 18/35 (51.6%) of the patients randomized to nasal-insertion, due to a lack of space or nasal polyps, compared to only one patient out of 31 (2.9%) that required a conversion from oral to nasal access due to a strong retching reflex. In another randomized trial, Choi et al. [31] enrolled 307 patients, with 17.7% of patients randomized to nasal access requiring a conversion to the oral route and no necessity of conversion from oral to nasal insertion reported. Bleeding at the insertion site was significantly more common in the nasal group (7% vs. 0%, *p* = 0.005), as well as coughing and shortness of breath. Nevertheless, willingness to return for future bronchoscopies if necessary was not significantly different between the two groups. The authors pointed out that the results of their studies might have been influenced by cultural and racial differences, as nasal cavity dimensions are usually lower in Asians compared to Caucasians.

The linear EBUS probe is larger and stiffer compared to the distal part of a conventional flexible bronchoscope. This led most interventional pulmonologists to perform EBUS-TBNA under general anesthesia through an endotracheal tube (or laryngeal mask) or under conscious sedation through the mouth. Under general anesthesia, the only possible accesses for EBUS scope are through the mouth or tracheostomy tube in tracheostomized patients, whereas under moderate sedation EBUS-TBNA can be executed through both oral and nasal access, as the safety and feasibility of this procedure through nasal access have been documented during the last decade [8]. A retrospective study by Beaudoin et al. [20] demonstrated that in 73.5% of the cases, EBUS-TBNA was possible through nasal access. In this case, there was no significant difference between nasal and oral routes regarding location and number of lymph nodes biopsied for each patient, procedure duration, and complications.

The same group [21] conducted a randomized controlled trial to compare patient’s perception of comfort during linear EBUS through the oral or nasal route under conscious sedation. No significant difference was observed in the primary outcome of patient comfort (8.3 vs. 8.4, *p* = 0.99), measured with a 10-point Likert scale, in the 220 patients that were randomized to the nasal or oral route (110 in each group) for EBUS-TBNA. Similarly, there was no significant difference in overall patient satisfaction and willingness to return, location of stations sampled, procedure duration, or total doses of sedatives and local anesthetics administered between the two groups. Twenty-seven out of 110 (24.5%) patients randomized to the nasal route had the procedure converted to the oral route due to failed nasal insertion, whereas no conversion from the oral to the nasal route was necessary. These results led the authors to conclude that for linear EBUS, the nasal and oral approaches confer a similarly high degree of patient comfort with similar complication rates and diagnostic yield. Thus, they infer that patient and physician preferences should dictate the route of insertion. A systematic literature review by Wahidi et al. [18] confirmed that there is little evidence to support the preferential use of one of the two accesses over the other one.

Notably, in 2020, Mittal et al. [22] described a case report of a patient who underwent esophageal endoscopic ultrasound using an endobronchial ultrasound bronchoscope (EUS-B) performed through nasal insertion as EBUS-TBNA was not feasible due to his tracheostomy tube size and complete vocal cord adduction. In this case, a conventional EUS-B was impossible due to a severely restricted mouth opening, so they performed a trans-nasal EUS-B. A subsequent prospective study [9] confirmed that EUS-B is feasible through the nasal route. In this study, nasal insertion was possible in 87.4% of the 119 patients enrolled, and the overall diagnostic yield of the EUS-B was 85.9%. The procedure was generally well tolerated, with a high willingness to return.

Our study had the purpose of investigating the use of nasal access for EBUS-TBNA in a third-level center, where these procedures are performed daily and where there is a relative shortage of anesthesiologists. For these reasons, the search for a technique assuring good patient compliance, better stability of the scope, and an improvement in the comfort of the bronchoscopist is highly appreciated. In our experience, the nasal access provides comparable comfort and, consequently, similar compliance for the patient. Furthermore, it gives the endoscopist better stability while holding the scope and avoids the problem caused by the annoying unintentional small movements that the patient makes with the tongue.

Our data suggest that the nasal route can be a valid alternative to EBUS-TBNA, in consideration of its safety and diagnostic accuracy. Of note, moderate sedation does not generally require the presence of an anesthesiologist, it carries fewer anesthetic-related complications, and it is cost-saving [5]. When the oral route is chosen, a coworker who manages the mouthpiece for bronchoscope insertion is useful while the first operator performs the bronchoscopy. Notably, the trans-nasal approach does not require this support.

In accordance with literature data, nasal access performed by the vast majority of the patients under moderate sedation has comparable diagnostic accuracy with procedures performed under deep sedation. On the other hand, interestingly, it has a similar safety profile compared with the procedure performed under anesthesiologic assistance and with the procedure performed via the oral route. In particular and a bit surprisingly, despite the fact that oxygen was administered through the nose during nasal access, there were no differences detected in terms of desaturations or the need for supplemental oxygen.

Nevertheless, it is important to say that in our experience, as shown in the data, deep sedation is applied in particular for longer procedures (i.e., for mediastinal systematic staging) or in patients who have previously undergone a biopsy. In both cases, as expected, we found a significant difference both for nasal and oral access compared with a laryngeal mask. Explaining the first situation, we prefer to perform mediastinal staging in deep sedation due to the longer expected procedural time, the fact that it frequently exceeds the relatively brief half-lives of the drugs pulmonologists use (midazolam, fentanyl, and meperidine), and the frequent need to view and sample minute nodes (usually smaller than 1 cm in diameter), for which it is preferable to avoid any interference such as the patients’ cough. The second circumstance reflects the fact that procedures under deep sedation with laryngeal mask access were adopted for mediastinal staging in patients who had been previously diagnosed with lung cancer and who had completed the staging with a systematic EBUS and in patients in whom a previous sampling attempt with the oral or nasal route had been tried without success due to patient intolerance. This latter was an infrequent event (two cases in the nasal access group and one case in the oral access group), but sufficient to determine its significance. 

Another important consideration looking at our results is that the nasal route is more frequently applied to male patients (and for this reason, “bigger” patients, considering height and weight), so it appears to be important to consider the physical features of the patients while attempting the nasal access.

There are several limitations to this study that need to be considered. First, it mainly does not let us discuss the real applicability of the nasal route in daily practice because it has not been systematically applied to all the patients. Second, some data are lacking, such as the duration of the procedures and the vital parameters recorded during the procedures themselves; these data could have been important for the identification of the real implications of the nasal route in the daily practice in terms of comfort and tolerability of the patients and gratification of the bronchoscopist. Third, a measure of the satisfaction of the patient could be crucial and is lacking in our work. On the other hand, we evaluated two major strengths of the work. First, the dimension of the casuistry and the fact that scarce data are available in the literature on this field. Second, the comparison not only between the nasal and the oral route for procedures performed mainly under moderate sedation, but also the presence of a comparison with a consistent number of procedures executed under deep sedation with an artificial airway such as the laryngeal mask.

In the evaluation of this technique, we consider important the role of a new prospective trial analyzing some fundamental aspects. Assuming that diagnostic accuracy and safety profiles are similar between the different access routes, first of all, the most important feature to evaluate is the comfort of the patients, because it implies better management of the sedative drugs and optimal collaboration during the exam. For this reason, questionnaires administered after the exam on the comfort felt during the procedure itself could be an affordable measure, while the magnitude of the change in vital signs could be an objective way to quantify the patient’s unconscious. On the other hand, also the satisfaction of the bronchoscopist could be important to measure because the better the endoscopist feels about the position of the scope and its handling, probably the fewer it would take to perform the procedure, consequently sparing time and resources, in addition to minimizing the onset of adverse events and reducing the dosage of sedative drugs. For this reason, the time of the procedure could be a good parameter in order to compare the different routes of access, even if there have not been identified differences in previous works.

In the future, the nasal route will probably be used more, considering the likely improvement in endoscope manufacturing and the further miniaturization of the sonographic probes. For this reason, maybe the nasal route will be the most frequent way of access adopted in the near future for EBUS-TBNA, as it is now for conventional flexible bronchoscopies.

## 5. Conclusions

In conclusion, our study highlights that nasal access for EBUS-TBNA has a comparable safety profile to the oral route and also to procedures performed under deep sedation with a laryngeal mask. Equally important, we report similar diagnostic accuracy in the procedure performed with this method of access in comparison to the others. Prospective studies are needed in order to evaluate the rate of successful usage of the nasal route when attempted, the satisfaction and comfort of the patient, as well as those of the bronchoscopist.

## Figures and Tables

**Figure 1 diagnostics-13-01405-f001:**
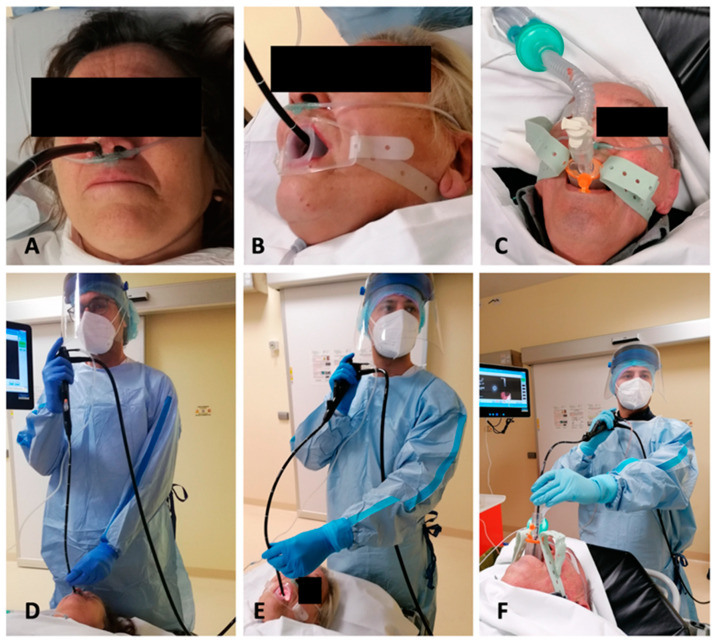
(**A**–**C**), entry ways of the EBUS scope: (**A**), nose; (**B**), mouth with mouthpiece; (**C**), laryngeal mask (i-gel). (**D**–**F**), endoscopist position in each of the entry ways: (**D**), nose; (**E**), mouth; (**F**), laryngeal mask. EBUS, endobronchial ultrasound.

**Figure 2 diagnostics-13-01405-f002:**
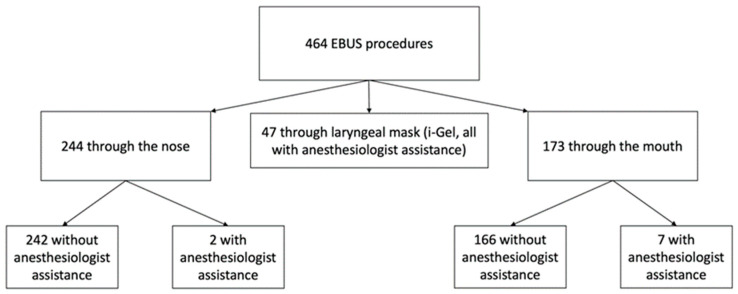
Division of the casuistry based on the entry way of the EBUS scope and anesthesiologist assistance. EBUS, endobronchial ultrasound.

**Table 1 diagnostics-13-01405-t001:** Inclusion and exclusion criteria for EBUS-TBNA, the endobronchial ultrasound-guided transbronchial needle aspiration.

Inclusion criteria: • clinical/radiological indication for EBUS-TBNA • willingness to undergo the planned procedure and subscription of a written informed consent • age > 18 years
Exclusion criteria: • procedures performed with the use of a rigid bronchoscope • intubated or tracheostomized patients • contraindications to bronchoscopy or sedation (e.g., severe hemodynamic instability) • antiplatelet agents (other than aspirin) or anticoagulants if not suspended • exams that included additional diagnostic procedures different from flexible bronchoscopy and EBUS-TBNA • procedures carried out by the resident physician

**Table 2 diagnostics-13-01405-t002:** Anagraphics, data regarding samples, and results in the overall population. SD, standard deviation; BMI, body mass index; NSCLC, non-small cell lung cancer; SCLC, small cell lung cancer.

	Overall Population (464)
Age (years, mean ± SD)	66.7 (±11.4)
Sex (males, %)	296 (63.8)
Death (%)	236 (50.9)
Weight (kg, mean ± SD)	73.5 (±16.3)
Height (cm, mean ± SD)	167 (±9.9)
BMI (kg/m^2^, mean ± SD)	26.2 (±4.9)
Previous biopsies (%)	81 (17.5)
Mediastinal staging (%)	136 (29.3)
Mediastinal adenopathies (%)	444 (95.7)
Adequate visualization (%)	455 (98.1)
Sample acquisition (%)	458 (98.7)
No. of stations sampled (mean ± SD)	1.7 (±1)
2R (%)	9 (1.9)
2L (%)	3 (0.6)
3 (%)	3 (0.6)
4R (%)	173 (37.3)
4L (%)	64 (13.8)
7 (%)	273 (58.8)
8 (%)	3 (0.6)
10R (%)	23 (4.9)
10L (%)	14 (3)
11R (%)	108 (23.3)
11L (%)	81 (17.5)
12R (%)	0 (0)
12L (%)	2 (0.4)
T (%)	30 (6.5)
No. of total needle passes (mean ± SD)	5.9 (±2.8)
Anesthesiologist assistance (%)	56 (12.1)
Diagnostic sample (%)	381 (82)
Chronic inflammation (%)	10 (2.2)
Anthracosis (%)	85 (18.3)
NSCLC (%)	177 (38.1)
SCLC (%)	40 (8.6)
Neuroendocrine large cell tumors (%)	1 (0.2)
Metastasis of other solid tumors (%)	19 (4.1)
Benign neoplasms (%)	1 (0.2)
Sarcoidosis (%)	37 (8)
Lymphoma (%)	10 (2.2)
Lymphocytes (%)	234 (50.4)
Suspicious for malignancy (%)	3 (0.6)
Inadequate sample (%)	24 (5.2)
False negative (%)	2 (0.4)

**Table 3 diagnostics-13-01405-t003:** Anagraphics, data regarding samples, procedural sedation, adverse events, and result comparisons between the nasal and oral access groups. * The mean, SD, and *p* values of propofol and remifentanil have been calculated, taking into account only cases with anesthesiologist assistance. SD, standard deviation; BMI, body mass index; NSCLC, non-small cell lung cancer; SCLC, small cell lung cancer.

	Nose (244)	Mouth (173)	*p* Value
Age (years, mean ± SD)	69.1 (±11.5)	68.3 (±11.4)	0.49
Sex (males, %)	169 (69.3)	96 (55.5)	0.005
Death (%)	123 (50.4)	94 (54.3)	0.49
Height (cm, mean ± SD)	168.1 (±10.3)	165.4 (±9.2)	0.005
Weight (kg, mean ± SD)	74.8 (±16.4)	71.7 (±16.8)	0.03
BMI (kg/m^2^, mean ± SD)	26.3 (±4.6)	26 (±5.4)	0.25
Previous biopsies (%)	37 (15.2)	29 (16.8)	0.68
Mediastinal staging (%)	74 (30.3)	37 (21.4)	0.04
Mediastinal adenopathies (%)	231 (94.7)	168 (97.1)	0.33
Adequate visualization (%)	238 (98.3)	170 (97.5)	0.74
Sample acquisition (%)	240 (98.4)	171 (98.8)	>0.9
No. of stations sampled (mean ± SD)	1.7 (±1.1)	1.6 (±0.9)	0.47
No. of total needle passes (mean ± SD)	5.9 (±2.9)	5.5 (±2.3)	0.1
Anesthesiologist assistance (%)	2 (0.8)	7 (4)	0.037
Sedative drugs			
Meperidine (mg, mean ± SD)	56.5 (±36.2)	50.1 (±36.3)	0.08
Midazolam (mg, mean ± SD)	5.9 (±2.7)	5.5 (±2.6)	0.14
Fentanyl (mg, mean ± SD)	0.044 (±0.059)	0.047 (±0.055)	0.58
Remifentanil (mg, mean ± SD) *	0 (±0)	0.36 (±0.75)	>0.9
Propofol (mg, mean ± SD) *	250 (±0)	454.3 (±296.6)	0.12
Adverse events (%)	25 (10.2)	17 (9.8)	>0.9
Epistaxis (%)	5 (0)	0 (0)	0.08
Mild airway bleeding (%)	11 (4.5)	1 (0.6)	0.017
Desaturation (%)	1 (0.4)	6 (3.5)	0.022
Hypertension (%)	1 (0.4)	1 (0.6)	>0.9
Use of sedative antagonists (%)	1 (0.4)	0 (0)	>0.9
Cough (%)	2 (0.8)	4 (2.3)	0.24
Agitation (%)	2 (0.8)	6 (3.4)	0.07
Stridor or bronchospasm (%)	3 (1.2)	4 (2.3)	0.45
Diagnostic sample (%)	205 (84)	142 (82)	0.59
Inadequate sample (%)	12 (4.9)	9 (5.2)	>0.9
False negative (%)	2 (0.8)	0 (0)	0.51

**Table 4 diagnostics-13-01405-t004:** Data regarding samples, procedural sedation, adverse events, and results of the three different access subgroups. * The mean, SD, and *p* values of propofol and remifentanil have been calculated, taking into account only cases with anesthesiologist assistance. SD, standard deviation; BMI, body mass index; NSCLC, non-small cell lung cancer; SCLC, small cell lung cancer.

	Nose (244)	Mouth (173)	Laryngeal Mask (47)	*p* Value
Previous biopsies (%)	37 (15.2)	29 (16.8)	15 (31.9)	0.02
Mediastinal staging (%)	74 (30.3)	37 (21.4)	25 (53.2)	0.0001
Mediastinal adenopathies (%)	231 (94.7)	168 (97.1)	45 (95.7)	0.5
Adequate visualization (%)	238 (98.3)	170 (97.5)	47 (100)	0.5
Sample acquisition (%)	240 (98.4)	171 (98.8)	47 (100)	0.7
No. of stations sampled (mean ± SD)	1.7 (±1.1)	1.6 (±0.9)	2.1 (±1.3)	0.02
No. of total needle passes (mean ± SD)	5.9 (±2.9)	5.5 (±2.3)	7.4 (±3.2)	0.0003
Anesthesiologist assistance (%)	2 (0.8)	7 (4)	47 (100)	<0.0001
Sedative drugs				
Meperidine (mg, mean ± SD)	56.5 (±36.2)	50.1 (±36.3)	0 (±0)	<0.0001
Midazolam (mg, mean ± SD)	5.9 (±2.7)	5.5 (±2.6)	1.3 (±2.1)	<0.0001
Fentanyl (mg, mean ± SD)	0.044 (±0.059)	0.047 (±0.055)	0.118 (±0.078)	<0.0001
Remifentanil (mg, mean ± SD) *	0 (±0)	0.36 (±0.75)	0.32 (±1)	0.64
Propofol (mg, mean ± SD) *	250 (±0)	454.3 (±296.6)	550.6 (±325.8)	0.13
Adverse events (%)	25 (10.2)	17 (9.8)	5 (10.6)	0.98
Epistaxis (%)	5 (0)	0 (0)	0 (0)	0.1
Mild airway bleeding (%)	11 (4.5)	1 (0.6)	2 (4.2)	0.06
Desaturation (%)	1 (0.4)	6 (3.5)	1 (2.1)	0.06
Hypertension (%)	1 (0.4)	1 (0.6)	0 (0)	0.9
Use of sedative antagonists (%)	1 (0.4)	0 (0)	0 (0)	0,6
Cough (%)	2 (0.8)	4 (2.3)	0 (0)	0.3
Agitation (%)	2 (0.8)	6 (3.4)	2 (4.2)	0.1
Stridor or bronchospasm (%)	3 (1.2)	4 (2.3)	0 (0)	0.4
Diagnostic sample (%)	205 (84)	142 (82)	34 (72)	0.16
Inadequate sample (%)	12 (4.9)	9 (5.2)	3 (6.4)	0.9
False negative (%)	2 (0.8)	0 (0)	0 (0)	0.4

## Data Availability

The data presented in this study are available upon request from the corresponding authors. The data are not publicly available due to the privacy policy for clinical information in Italy.

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
