# Peer review of "Safety and Diagnostic Accuracy of the Transnasal Approach for Endobronchial Ultrasound-Guided Transbronchial Needle Aspiration (EBUS-TBNA)"

_diagnostics, 2023, doi:10.3390/diagnostics13081405_

Round 1

Reviewer 1 Report

Dear authors, 

The paper described a very well know and already proven EBUS technique. As you mentioned, there was a randomized controlled study and many retrospective study investigating the same topic of nasal EBUS. I personally know many centers who are doing nasal EBUS and did thousands over the last 15 years. 

You cannot say you are studying the feasibility of a technique that was proven feasible many times including in Randomized controlled study. 

Also, Please consider the following:

1. add recent history of epistaxis to the exclusion criteria. Maybe in the last year or 2 years. 

2. Line 107-109: Please describe the nasal approach clearer. It is hard to understand the way you are describing. You can clearly say something like : The EBUS scope was introduced through the open nostril after using lubricating jelly. The scope was then gently introduced through the middle and lower nasal Choana, without resistance  and trying to avoid trauma of the nasal walls. 

3. line 120: upper paratracheal ones. 

4. Line 120-121: the sentence “with laryngeal mask…..airway management should be removed. 

5. in line 138: Lidocaine 300 mg in each nostril? then how many on the larynx, trachea, bronchi? that would be an overdose !!! please clarify how many mg per kg approximately was given totally. 

6. In line 143: shouldn’t the patient be asked which nostril is open and most patient will tell you which one is more open that’s how the bronchoscopist should consider which nostril to go through first. 

7. except for tamponade: how did you tamponade? please specify

8. in line 223: Intravenous steroids? why would you give that for? and how did that resolve the laryngospasm. Takes hours to start to work. Please explain based on what data did you treat acute laryngospasm with IV steroids and resulted in rapid recovery?

9. in line 294: by mouth or tracheostomy tube in patient who are trached.

Thank you 

Author Response

RESPONSE TO REVIEWER 1 COMMENTS

Dear authors, The paper described a very well know and already proven EBUS technique. As you mentioned, there was a randomized controlled study and many retrospective study investigating the same topic of nasal EBUS. I personally know many centers who are doing nasal EBUS and did thousands over the last 15 years.  You cannot say you are studying the feasibility of a technique that was proven feasible many times including in Randomized controlled study.

RESPONSE: We thank the Reviewer for this consideration. As she/he underlines, to date only one randomized trial has been conducted (Bodoin et al, J Bronchology Interv Pulmonol, 2016); it is an interesting and well-conduced study, in which 110 subjects underwent the nasal and 110 the oral route. Apart from that, scientific literature in this field is scarce and it involves a retrospective study on 209 patients (Bodoin et al, Lung, 2014) and some case reports. Our retrospective study described 464 EBUS-TBNA: it compares the nasal route to the oral one and to the use of laryngeal mask. We think that our data is relevant, enhances the knowledge in the field, supports the centres that use the nasal approach, and encourages the other professionals to consider this route. We agree with the Reviewer that the feasibility of this technique was already described in the mentioned articles, so we modified our manuscript to describe this aspect differently. Thank you very much for the suggestion.

Also, Please consider the following:

  1. add recent history of epistaxis to the exclusion criteria. Maybe in the last year or 2 years.

RESPONSE: A history of epistaxis was not an exclusion criterion for our study but as a matter of fact it was taken into consideration by the interventional pulmonologist in the choice of the route. We added a phrase about it in lines 154-156. We thank the Reviewer for the comment.

  1. Line 107-109: Please describe the nasal approach clearer. It is hard to understand the way you are describing. You can clearly say something like : The EBUS scope was introduced through the open nostril after using lubricating jelly. The scope was then gently introduced through the middle and lower nasal Choana, without resistance and trying to avoid trauma of the nasal walls.

RESPONSE: We thank the Reviewer for the suggestion; accordingly, we emended the paragraph.

  1. line 120: upper paratracheal ones.

RESPONSE: We thank the Reviewer: we corrected the error.

  1. Line 120-121: the sentence “with laryngeal mask…..airway management should be removed.

RESPONSE: We thank the Reviewer for her/his proposal; accordingly, we removed the sentence.

  1. in line 138: Lidocaine 300 mg in each nostril? then how many on the larynx, trachea, bronchi? that would be an overdose !!! please clarify how many mg per kg approximately was given totally.

RESPONSE: We thank the Reviewer for the correction and we apologize for the mistake contained in this sentence. The lowest dose of lidocaine ensuring good bronchoscopic conditions and patient comfort was used: approximately, we used 4-6 mg/kg of lidocaine. We emended the paragraph.

  1. In line 143: shouldn’t the patient be asked which nostril is open and most patient will tell you which one is more open that’s how the bronchoscopist should consider which nostril to go through first.

RESPONSE: We thank the Reviewer for the hint, which effectively reflects a question frequently asked by the bronchoscopist to the patient and, other times, reported directly by the patient before the beginning of the exam. We add this occurrence at lines 149-151.

  1. except for tamponade: how did you tamponade? please specify

RESPONSE: We thank the Reviewer for the advice. Tamponade is defined by the placement of a small sterile gauze pad in the nostril where the epistaxis was observed. We added this description in lines 232-233.

  1. in line 223: Intravenous steroids? why would you give that for? and how did that resolve the laryngospasm. Takes hours to start to work. Please explain based on what data did you treat acute laryngospasm with IV steroids and resulted in rapid recovery?

RESPONSE: We thank the Reviewer for the correction. Effectively, we have improperly referred to a sign observed as a complication during some procedures as “laryngospasm”, while it is more correct to refer as “stridor”, which can arise from a glottic or subglottic oedema, described following intubation or extubation but potentially seen also during bronchoscopy due to the stimulation of the larynx and first third of thrachea. For this clinical situation, the use of steroids is debated but accepted and described in literature (Khemani RG, Randolph A, Markovitz B. Corticosteroids for the prevention and treatment of post-extubation stridor in neonates, children and adults. Cochrane Database Syst Rev. 2009 Jul 8;2009(3):CD001000. doi: 10.1002/14651858.CD001000.pub3). The mistake is linguistic and linked to the habit, during endoscopic procedures, of referring to the objective sign of the presence of stridor as laryngospasm. We apologize for the misunderstanding. We have modified this wording in the text specifying also the use of aerosol therapy for the complications of bronchospasm and stridor.

  1. in line 294: by mouth or tracheostomy tube in patient who are trached.

RESPONSE: We thank the Reviewer. We modified the sentence.

Thank you

Reviewer 2 Report

This study retrospectively compared oral and nasal insertion when inserting EBUS-TBNA. The paper is well written, and although there are some limitations, I think it is well considered.

Author Response

RESPONSE TO REVIEWER 2 COMMENTS

This study retrospectively compared oral and nasal insertion when inserting EBUS-TBNA. The paper is well written, and although there are some limitations, I think it is well considered.

RESPONSE: We thank the Reviewer for her/his very positive words. We are very pleased that she/he considered the paper well written and worthy of consideration.

Reviewer 3 Report

I read manuscript titled "Safety and diagnostic accuracy of the trans nasal approach for endobronchial ultrasound-guided transbronchial needle aspiration (EBUS-TBNA)" with great interest.

This is an important issue to address in everyday routine practical use of EBUS-TBNA.

I do however have a minor suggestion: please rewrite the inclusion criteria; it should not state that patients must be from particular hospital and particular time frame. The inclusion criteria should be more open, e.g.

1. clinical/radiological indication for EBUS TBNA,

2. Absence of contraindications to bronchoscopy (or analgesic sedation)

3. Willingness to undergo planed procedure (institutional ICF) ... etc.

I advise to change the exclusion criteria too.

I do not have any additional issues with the paper and after the correction I would suggest rapid publication.

God speed

The reviewer

Author Response

RESPONE TO REVIEWER 3 COMMENTS

I read manuscript titled "Safety and diagnostic accuracy of the trans nasal approach for endobronchial ultrasound-guided transbronchial needle aspiration (EBUS-TBNA)" with great interest.

This is an important issue to address in everyday routine practical use of EBUS-TBNA.

RESPONSE: We thank the Reviewer for her/his enthusiastic words. We are very pleased that she/he considered the paper very interesting, and she/he judged the argument important to address the routine use of EBUS-TBNA.

I do however have a minor suggestion: please rewrite the inclusion criteria; it should not state that patients must be from particular hospital and particular time frame. The inclusion criteria should be more open, e.g.

  1. clinical/radiological indication for EBUS TBNA,
  2. Absence of contraindications to bronchoscopy (or analgesic sedation)
  3. Willingness to undergo planed procedure (institutional ICF) ... etc.

I advise to change the exclusion criteria too.

RESPONSE: We thank the Reviewer for her/his good suggestions. We modified the paper (including Table 1) accordingly with them.

I do not have any additional issues with the paper and after the correction I would suggest rapid publication. God speed The reviewer

RESPONSE: We thank the Reviewer for helping us to improve the paper and for recommending a rapid publication.

Round 2

Reviewer 1 Report

All concerns were addressed properly. Thank you